# Investigating the Role of Antibiotics on Induction, Inhibition and Eradication of Biofilms of Poultry Associated *Escherichia coli* Isolated from Retail Chicken Meat

**DOI:** 10.3390/antibiotics11111663

**Published:** 2022-11-19

**Authors:** Aisha Noreen, Hamid Masood, Jaweria Zaib, Zara Rafaque, Areeta Fatima, Hira Shabbir, Javaria Alam, Aisha Habib, Saba Noor, Kinza Dil, Javid Iqbal Dasti

**Affiliations:** 1Lab of Microbial Genomics and Epidemiology, Department of Microbiology, Quaid-i-Azam University, Islamabad 45320, Pakistan; 2Department of Microbiology, Hazara University, Mansehra 21120, Pakistan

**Keywords:** biofilm, colistin, multidrug resistance, virulence, *E. coli*, sub-MIC

## Abstract

Background: Widespread use of antibiotics as growth promoters and prophylactic agents has dramatic consequences for the development of antibiotic resistance. In this study, we investigated effects of selected antibiotics on bacterial biofilms and performed extensive antibiotic and VF profiling of poultry-meat associated *E. coli* strains. Methods: Antibiotic susceptibility was performed by a disc diffusion method, followed by molecular screening of resistance and virulence determinants. Further biofilm formation assays, MIC-p, MIC-b, MBIC and MBEC, were performed using standard tissue culture plate method. Results: In total, 83 (75%) samples were confirmed as *E. coli* from poultry sources, 26 different antibiotics were tested, and maximum numbers of the isolates were resistant to lincomycin (100%), while the least resistance was seen against cefotaxime (1%) and polymyxin B (1%). Overall, 48% of the isolates were ESBL producers and 40% showed carbapenemase activity; important virulence genes were detected in following percentages: *fimH*32 (39%), *papC*21 (25%), *iutA*34 (41%), *kpsMT*-II23 (28%), *papEF*9 (11%), *papGII*22 (27%) and *fyuA*13 (16%). Colistin showed remarkable anti-biofilm activity, while at sub-MIC levels, gentamicin, ceftriaxone and enrofloxin significantly (*p *< 0.01) inhibited the biofilms. A strong induction of bacterial biofilm, after exposure to sub-minimal levels of colistin clearly indicates risk of bacterial overgrowth in a farm environment, while use of colistin aggravates the risk of emergence of colistin resistant *Enterobacteriaceae*, a highly undesirable public health scenario.

## 1. Introduction

*Escherichia coli* (*E. coli*), a Gram-negative rod-shaped bacterium belongs to the family Enterobacteriaceae. As a part of human microbiota, it aids in digestion process and synthesis of vitamins. *E. coli* is one of the first microorganisms that colonize infant gut after birth [1]. Pathogenic *E. coli* equipped with diverse virulence factors cause different infections, including urinary tract infections, hospital-acquired pneumonia, GI tract infections and meningitis [2]. Based on site of infection, bacteria are categorized into two major sub-groups; intestinal and extra-intestinal pathogenic *E. coli.* ExPEC include uropathogenic, septicemia-associated, meningitis-associated and avian pathogenic strains. Isolates from different environmental samples such as sewage, food, water, domestic and wild animals show remarkable resemblance to ExPEC, causing human infections [3]. Recently, presence of ExPEC strains was confirmed in various food products [4,5]. Meat has been used for centuries as a source of food, and chicken meat is the second most-consumed meat type in the world [6,7]. Despite COVID-19, in year 2020 there was an estimated increase of 9.1% in poultry meat production in Pakistan [8]. Apart from poultry meat contamination, as a food-bore pathogen, *E. coli* causes colibacillosis, septicemia, entero-colitis and omphalitis in the poultry [9]. Antibiotics such as penicillin, salinomycin, chlortetracycline, bacitracin, and colistin are widely used for the growth promotion, prophylaxis and treatment of infectious diseases [10]. Particularly in Pakistan, neomycin, amoxicillin, enrofloxacin, colistin, doxycycline, and tylosin are frequently used as prophylactic agents in the poultry industry. Presence of multidrug-resistant strains of *E. coli* in poultry has been reported in different parts of the world [11]. Earlier, Olsen and co-workers attributed 50% of the layer flock mortalities to antibiotic-resistant *E. coli* [12]. The most common antibiotic resistance phenomenon among Enterobacteriaceae is the production of extended spectrum β-lactamases (ESBLs). These genes located on plasmids are transferred horizontally from one bacterial cell to another and foster dissemination of antibiotic-resistance at an exponential rate [13]. For the human host, contaminated meat can be a source of *E. coli* infection and direct transmission occurs via contaminated surfaces. Moreover, avian pathogenic *E. coli* is able to stabilize itself on meat processing equipment by forming biofilms [14]. Biofilm formation requires set of virulence factors needed for the adherence and colonization of different surfaces [15]. Bacteria as biofilms show remarkable tolerance towards antibiotics [16,17]. Studies have been reported on the effects of sub-minimal inhibitory concentrations (sub-MIC) of antibiotics on *E. coli*, confirming inhibitory or stimulatory effects on the biofilms [16,17]. Due to their widespread association with human health and food chain, understanding current antibiotic resistance patterns and nature of the prevailing *E. coli* strains in terms of their virulence potential is crucial. Moreover, use of antibiotics in farm environment, even at sub-minimal concentrations leads to persistence of bacteria as biofilm that serves as an important reservoir for the development of MDR strains. Such strains are closely linked to the food chain; therefore transmissibility to humans is inevitable. In this context, the present study was conducted to investigate effects of ceftriaxone, gentamicin, enrofloxacin and colistin on the biofilm formation of *E. coli* recovered from poultry meat samples.

## 2. Results

### 2.1. Prevalence of Antibiotic Resistance among E. coli

In total, 83 (75%) of the isolates were confirmed as *E. coli* from poultry sources. The antibiotic susceptibility testing was performed for 26 different antibiotics. Higher resistance was observed against lincomycin (100%), oxytetracycline (90%), ampicillin (96%), amoxicillin (90%), tetracycline (90%), streptomycin (81%), trimethoprim (77%), chloramphenicol (64%), levofloxacin (61%), ciprofloxacin (61%), doxycycline (60%), neomycin (53%), cephalothin (43%) and colistin (polymyxin E, 26%). Lower resistance was observed against cefotaxime (1%), polymyxin B (1%), augmentin (5%), cefepime (5%), imipenem (5%), tobramycin (6%), cefixime (7%), ceftazidime (7%), gentamicin (8%), nitrofurantoin (8%) and meropenem (9%) (*p* > 0.05). Overall, *n* = 40 (48%) of the isolates were confirmed as ESBL producers. Detailed comparison of antibiotic resistance traits of MDR, XDR, ESBL and non ESBL strains are shown in Table 1 and Table 2.

### 2.2. Detection of VF and ESBL Genes

Overall, 41% of the isolates carried *iutA*, 39% *fimH*, 28% *kpsMTII*, 27% *papGII* and 25% *papC*. Likewise, *fyuA* and *papEF* were detected in 16% and 11% of the isolates, respectively. In ESBL producing isolates, *bla-*_TEM_ was the most frequently detected ESBL factor (55%), followed by *bla-*_OXA_ (20%) and *bla*_-SHV_ (10%). A significant association (*p* < 0.05) was observed between ESBL phenotypes and occurrence of following VF genes; *iutA*, *kpsMTII* and *fyuA*. Genotypic and phenotypic correlation of ESBL is shown in (Table 2).

### 2.3. Biofilm Formation Potential

Out of a total *n* = 83, *E. coli* isolates, 29% showed strong biofilm formation, while 37% formed biofilms at moderate levels and 34% were weak biofilm formers.

### 2.4. Measurement of MIC-p and MIC-b

The MIC-p and MIC-b values for ceftriaxone, colistin, enrofloxacin and gentamicin were determined. For ceftriaxone and enrofloxacin, MIC-p was ≤0.5 µg/mL, while for colistin and gentamicin, it was ≥1 µg/mL. MIC-b values of ceftriaxone ranged from 16 µg/mL to 32 µg/mL. Likewise, for enrofloxacin and colistin, it was 4 µg/mL and ≥1 µg/mL, respectively (Table 3).

### 2.5. Determination of MBIC/MRC and MBEC

MBIC/MRC for the tested strains (*n* = 2), #7 and #21, were determined as follows: gentamicin (8 µg/mL and 32 µg/mL), colistin (8 µg/mL and 16 µg/mL), ceftriaxone (128 µg/mL and 256 µg/mL) and enrofloxacin (256 µg/mL and 64 µg/mL). Likewise, MBEC values for the gentamicin (32 µg/mL and 128 µg/mL), colistin (128 µg/mL and 64 µg/mL), ceftriaxone (>2048 µg/mL) and enrofloxacin (>2048 µg/mL) were recorded. Although MBEC of ceftriaxone and enrofloxacin (>2048 µg/mL) were higher than colistin and gentamicin (ranging from 32 µg/mL to 128 µg/mL), both antibiotics (gentamicin and colistin) very effectively eradicated biofilms of selected *E. coli* isolates (Table 4).

### 2.6. Effect of Sub-MICs on Biofilm 

The effect of sub-MIC of colistin on biofilm was investigated at different time intervals (after, 4 h, 8 h, 12 h and 24 h) and showed variable reduction at different concentrations. Similarly, ceftriaxone, enrofloxacin and gentamicin also showed variable reductions at different concentrations at different time intervals. Overall, colistin showed stimulatory effects on biofilm formation, specifically after 24 h of incubation at sub-MIC level. In contrast, gentamicin, ceftriaxone and enrofloxacin showed inhibitory effects at sub-MIC level (Figure 1, Figure 2, Figure 3, Figure 4, Figure 5, Figure 6, Figure 7 and Figure 8).

## 3. Discussion

*E. coli* is one of the frequently encountered bacteria in poultry meat. In this study, isolation rate of *E. coli* from chicken meat was 75%. Earlier, varying resistance patterns against antibiotics have been reported from this region [18,19]. In different geographical areas different antibiotics have been used as feed additives, and prophylactic agents; consequently frequent use of these agents in farm environment asserts significant selective pressure for the antibiotic resistance development. For example, lincomycin hydrochloride soluble powder has been widely used for treating necrotic enteritis caused by *Clostridium perfringens* in different geographical areas, and chronic respiratory diseases caused by mycoplasma and respiratory bacterial infections were widely treated with the same antibiotic. On other hand, lincomycin has frequently been used in poultry feed (29.09 mg/fPU) in Pakistan [20,21]. In this study, 100% of the isolates were resistant to Lincomycin. Likewise, ampicillin is used in poultry to treat different bacterial infections caused by *E. coli*, Salmonella and Clostridia spp. and it has been one of the highly recommended therapeutic treatment options for the infections caused by Clostridia spp. Currently, due to loss of efficacy, it’s not been recommended in this region [21]. In this study, 96% of the *E. coli* isolates were resistant to ampicillin. Another broad-spectrum antibiotic amoxicillin is used to treat fowl typhoid, colibacillosis, and necrotic enteritis. In this study, resistance against amoxicillin was 90%. For the treatment of urinary tract infections caused by ExPEC (UPEC), fluoroquinolones are frequently prescribed in Asia, compounding 24% of the antibiotic prescription and are enlisted first choice for pyelonephritis [22,23,24]. In this study, 61% of the *E. coli* isolates were resistant to ciprofloxacin and levofloxacin. Surprisingly, even today, in Pakistan, doxycycline is frequently used in poultry, particularly in combination with tylosin and colistin to treat CRD, while in India, this antibiotic is banned in food producing animals and poultry to curb the AMR challenge [25]. Similarly, China already banned the use of colistin in poultry. In this study, we observed 27% of the isolates were resistant to colistin. Recently, we reported combined resistance to carbapenem and colistin in *Klebsiella pneumoniae isolates* that was linked to the presence of *mcr-1* and *mcr-2*genes [26]. Here, we reiterate that colistin is one of the highest priority critically important antibiotics (HPCIA) enlisted by WHO; therefore, it is crucial to completely ban its use for the growth promotion in poultry industry, particularly to limit co-selection of carbapenem and colistin resistant *Enterobacteriaceae*. Cationic polymyxins, cyclic lipodecapeptide antibiotics, interact with lipid A part of LPS that contains anionic phosphate and pyrophosphate groups, which bind with polymyxin, eventually leading to the release of LPS and loss of function for Gram-negative cells [27]. Polymyxins confer rapid bactericidal actions and are postulated in inhibition of biofilm formation of *A. baumannii* and UPEC indicating its suitability to treat biofilm associated infections [28,29]. In contrast, however, induction of biofilm in different bacterial species upon exposure to sub-MIC levels of other antibiotics has also been reported [17,30,31]. Since in this study, we observed 27% of the *E. coli* isolates resistant to polymyxin E (colistin), obviously a much higher prevalence rate when compared to UPEC and *K. pneumoniae*. 

We therefore hypothesized, role of polymyxins and other antibiotics in induction, inhibition and eradication of biofilms of poultry associated *E. coli*. Hence, we tested four frequently used antibiotics in poultry farms, colistin, gentamicin, enrofloxacin and ceftriaxone. Our results confirmed biofilm formation as early as 4 h of incubation and the highest level of biofilm formation was observed after 24 h of incubation. In biofilm form cells otherwise susceptible to all four antibiotics, became tolerant to significant levels of ceftriaxone and enrofloxacin (~32–64 times), when compared to their planktonic forms (MIC-p). Quite interestingly, our results showed lower MIC-b in comparison to MIC-p for other two antibiotics, colistin and gentamicin. Earlier, Klinger et al. (2017) reported inhibition of biofilm formation in colistin resistant isolates [29,32]. In contrast to that study, we used colistin susceptible strong biofilm former isolates of *E. coli*, definitely more relevant for such assessments. Taken together, our results regarding MBIC/MRC and MBEC confirm higher efficacy of colistin on biofilm inhibition and eradication. We also tested all four antibiotics for their efficacy on biofilm forming ability of selected isolates at sub-MIC level. Though there was variation in biofilm inhibition at 4–12 h, sub-MIC of gentamicin, ceftriaxone and enrofloxin resulted into significant (*p* < 0.01) reduction of biofilms. However, on contrast a significant induction and increase in biofilm formation was observed upon exposure to sub-MIC level of colistin, after 24 h of incubation. A recent study on *Acinetobacter baumannii* reported similar results, where a notable increase in the biofilm was observed in the presence of sub-MIC of colistin [33]. Colistin has been used in poultry industry as a growth promoter and to prevent *Enterobacteriaceae* mediated infections, including APEC. Recently, emergence of plasmid mediated colistin resistance in China was reported [34]. Exposure to sub-optimal concentration usually falls within/below the selection window (MSW) that eventually selects the resistant phenotypes, which, in turn, enhances the biofilm and virulence potential of pathogens. On a molecular level, sub-MIC of colistin apparently induces the expression of biofilm associated genes such as poly-acetyl-glucosamine-porin (*pgaA*), autoinducer synthase(*abaI*) and efflux pumps (*adeB and adeG*) [32,33].

Henceforth, it is very important to stop the irrational use of this antibiotic as prophylactic in farm practice, as even at sub-MIC level of antibiotic may lead to the dramatic surge in colistin resistance *Enterobacteriaceae* and widespread dissemination of plasmid encoded *mcr genes.*

## 4. Materials and Methods

### 4.1. Isolation and Identification of Bacterial Samples

A total of 110 meat samples were collected randomly from retail chicken meat shops of Peshawar, Pakistan (34°0’28” N, 71°34’24” E). The study was carried out from February 2018 to December 2020 at the Department of Microbiology, Quaid-i-Azam University, Islamabad, Pakistan. Meat samples were collected in sterile falcon tubes and were transported to the laboratory in a cool bank within 24 h of collection. These frozen meat samples were thawed at room temperature and transferred to Brain Heart Infusion (BHI) broth in sterile falcon tubes. The broth was incubated at 37 °C for 24 h to promote growth of *E. coli*. After 24 h, 500 μL of broth was separately transferred to freshly prepared MacConkey agar (Oxoid Ltd., Hamsphire, UK), and plates were subsequently incubated at 37 °C for 24 h. After the visual inspection for colony morphology, Gram-staining was performed to carry out microscopic examination. In total, 83 isolates were identified as *E. coli* by using standard biochemical and molecular methods as described elsewhere [35]. Bacterial samples were cultured on MacConkey agar and stored in glycerol stocks at −20 °C. 

### 4.2. Antibiotic Susceptibility Testing

Antibiotic susceptibility testing and phenotypic detection of ESBL production was performed as per CLSI guidelines [36]. MDR and XDR strains were defined as described elsewhere [37]. Bacterial isolates were tested against twenty-six different antibiotics (Oxoid) commonly used in public health and livestock sectors (Table 1). Briefly, 0.5 McFarland bacterial suspensions were prepared and using a sterile cotton swab they were spread on plates containing Muller–Hinton Agar (MHA). Antibiotic discs of known concentrations were placed on plate using sterile forceps. Plates were incubated at 37 °C for 16–18 h and zone diameters were measured.

### 4.3. PCR Detection of Virulence and ESBL Genes

Detection of eight virulence genes (*fimH*, *papC*, *iutA*, *kpsMT-II, papEF, papGII*, *fyuA* and *papGIII*) and four ESBL genes (*bla-*_TEM_, *bla-*_OXA_, *bla*-_SHV_ and *bla*-_PSE_) was carried out by polymerase chain reaction as described elsewhere [33]. Bacterial DNA was extracted by using phenol–chloroform method. PCR conditions were as follows: 95 °C for 1 min, followed by 35 cycles of denaturation at 95 °C for 45 s, annealing at 56 °C for 45 s, extension at 72 °C for one minute and a final extension at 72 °C for 10 min. The amplified products were observed on agarose gel and band sizes were compared with DNA ladder of 100 bp (Solis Biodyne).

### 4.4. Biofilm Formation Assay

A total of 83 *E. coli* isolates were tested for biofilm formation. Quantitative assessment of biofilms was performed by using microtiter plate method, a gold standard for biofilm quantification. Biofilm quantification was performed as previously described [17]. Briefly, a 200 µL of standardized 0.5 McFarland suspension was added to a 96 well clear, sterile polystyrene microtiter plate, and incubated at 37 °C for 24 h. After the incubation, content was removed, and the cells were washed three times with 0.9% phosphate buffer saline (PBS). Adherent bacteria were stained with crystal violet for 15 min and wells were again washed with PBS. Each sample was processed in triplicate and *E. coli* strain ATCC25922 was used as an experimental control. Wells without bacterial suspension (sterile MH broth) were taken as negative control. 

### 4.5. Measurement of MIC-p and MBC

Two strong biofilm forming isolates (isolate #7 and #21) sensitive to all four antibiotics colistin, ceftriaxone, enrofloxacin and gentamicin were selected. To confirm sensitivity, minimum inhibitory concentrations (MIC) against colistin, ceftriaxone, enrofloxacin and gentamicin were measured by following standard procedure [36]. After the incubation (24 h at 37 °C), starting from lowest concentration, the first dilution of the tested antibiotic showing no turbidity was taken as the MIC of that particular antibiotic. The clear well next to the MIC well was tested for Minimum Bactericidal Concentration (MBC) by plating the contents of the well on the nutrient agar plate. Absence of growth on plate after 24 h incubation at 37 °C was confirmed as MBC.

### 4.6. Determination of MIC-b, MBIC/MRC and MBEC

The two selected isolates (isolate #7 and #21) were then tested for Minimum Inhibitory Concentration of biofilms (MIC-b), Minimum Biofilm Inhibition Concentration (MBIC/MRC) and Minimum Biofilm Eradication Concentration (MBEC) of colistin, ceftriaxone, enrofloxacin and gentamicin through procedures described elsewhere [17]. 

### 4.7. Sub-MIC Concentration of Ceftriaxone

The effect of sub-MIC concentrations of ceftriaxone, gentamicin, enrofloxacin and colistin on biofilm forming ability of ExPEC was tested at different time intervals, i.e., 4 h, 8 h, 12 h and 24 h. The antibiotics were diluted at sub-MIC levels of 0.5 µg/mL 0.25 µg/mL, 0.125 µg/mL and 0.0625 µg/mL; MH broth was used for the preparation of antibiotic dilutions. For sub-MIC level testing, a 100 µL of each diluted antibiotic was added to each treatment well containing a 100 µL of standardized bacterial inoculum. All micro-titer plates were incubated at 37 °C. Quantification of biofilms was performed by measuring OD at 540 nm after adding crystal violet dye [17]. For all assays, antibiotic free standardized bacterial suspension was added as a positive control while bacteria free antibiotic solution was included as negative control. Biofilm formation was analyzed by comparing the reading with OD cut-off values.

Cut off OD of negative control (NC) was determined by using formula: ODcut off = ODavg of NC + 3 × standard deviation (SD) of ODs of NC. Each isolate was then categorized according to the following criteria: weak biofilm producer, OD = 2 × ODc; moderate biofilm producer, 2 × ODc ≤ 4 × ODc; strong biofilm producer, OD ≥ 4 × ODc. * The cut off OD for NC was 0.055.

### 4.8. Statistical Analysis

All the biofilm assays were performed in triplicates and repeated at least twice to ensure reproducibility of results. Statistical analysis of antibiotic resistance, virulence factors, resistance factors and that of multi-/extensively drug resistant strains (MDR/XDR) was performed by chi-square test using GraphPad Prism v.7.04. A *p*-value < 0.05 was considered statistically significant. Differences in biofilm formation upon treatments were concluded by using paired *t*-test.

## 5. Conclusions

Conclusively, irrational use of colistin as a prophylactic agent in farm practice results in to dramatic surge of resistant strains of *Enterobacteriaceae* linked to the human food chain and public health. Henceforth, this vital antibiotic must be restricted for rational use only.

## Figures and Tables

**Figure 1 antibiotics-11-01663-f001:**
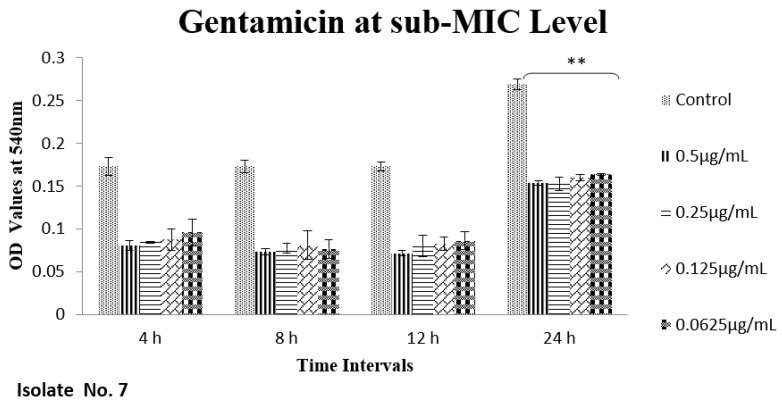
Shown are the effects of gentamicin on *E. coli* biofilms (OD 540 nm). Bar graphs showing significant reduction (** *p* value < 0.01) in biofilm formation at different time points when compared with the control samples.

**Figure 2 antibiotics-11-01663-f002:**
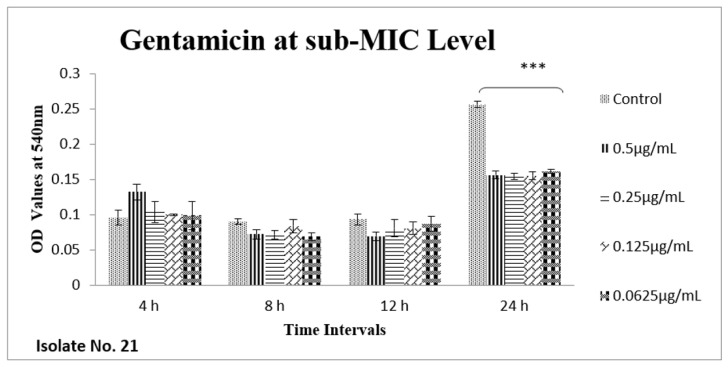
Bar graph shows the effects of gentamicin on *E. coli* biofilms (OD 540 nm). Overall significant reduction (*** *p* value < 0.001) in biofilm formation was observed after 24 h in comparison to the control samples.

**Figure 3 antibiotics-11-01663-f003:**
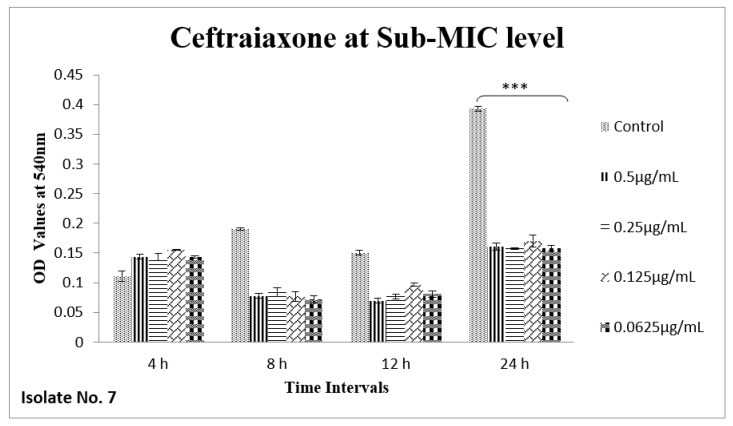
Shown are the effects of ceftriaxone on biofilm formation of *E. coli* at different time intervals (OD 540 nm). Bar graph represents significant reduction (*** *p* value < 0.001) of biofilm formation at different time points compared to the control samples.

**Figure 4 antibiotics-11-01663-f004:**
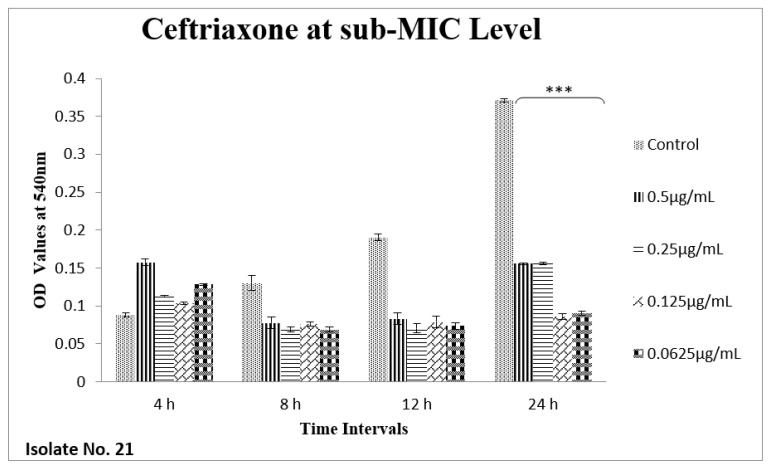
Shown are the effects of ceftriaxone on *E. coli* biofilms at different time intervals (OD 540 nm). Overall significant reduction (*** *p* value < 0.001) in biofilm formation was observed up to 24 h, when compared to the control samples.

**Figure 5 antibiotics-11-01663-f005:**
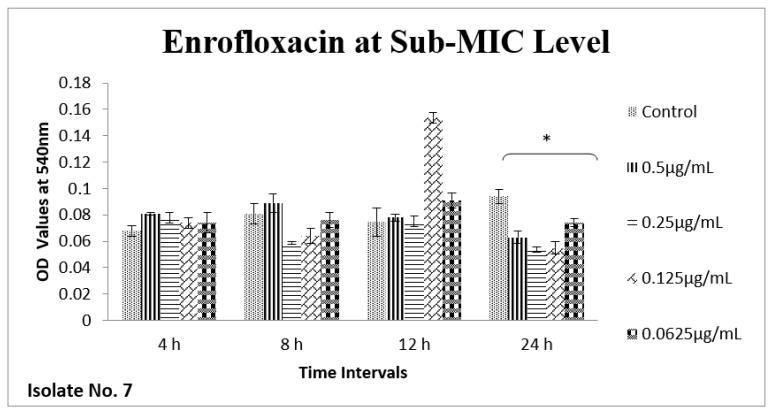
Shown are the effects of enrofloxacin on bacterial biofilm at different time points (OD 540 nm). Bar graph represents significant reduction (* *p* value < 0.05) in biofilm formation after 24 h of incubation when compared to the control samples.

**Figure 6 antibiotics-11-01663-f006:**
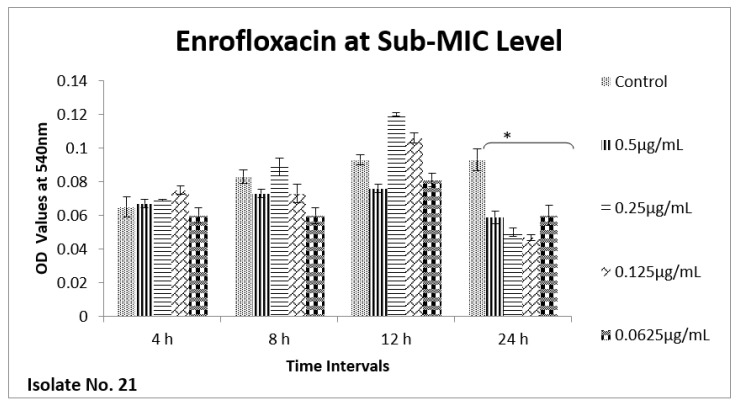
Shown are the effects of enrofloxacin on *E. coli* biofilm at different time intervals (OD 540 nm). Bar graph represents significant reduction (* *p* value < 0.05) in biofilm formation up to 24 h, of incubation when compared to the control samples.

**Figure 7 antibiotics-11-01663-f007:**
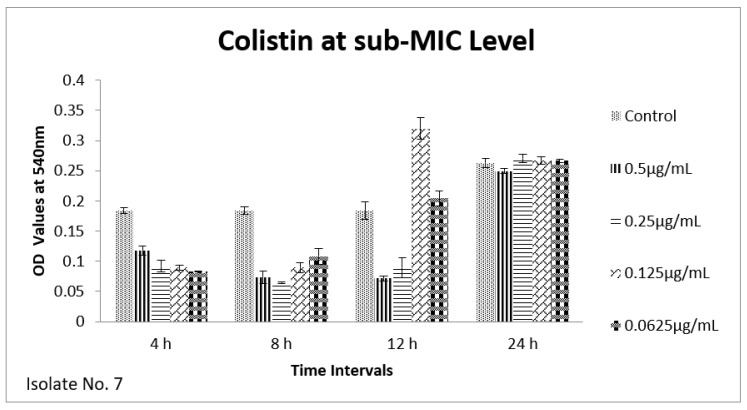
Shown are the effects of colistin on *E. coli* biofilm at different time points (OD 540 nm). Bar graph represents highest biofilm formation at 24 h of incubation.

**Figure 8 antibiotics-11-01663-f008:**
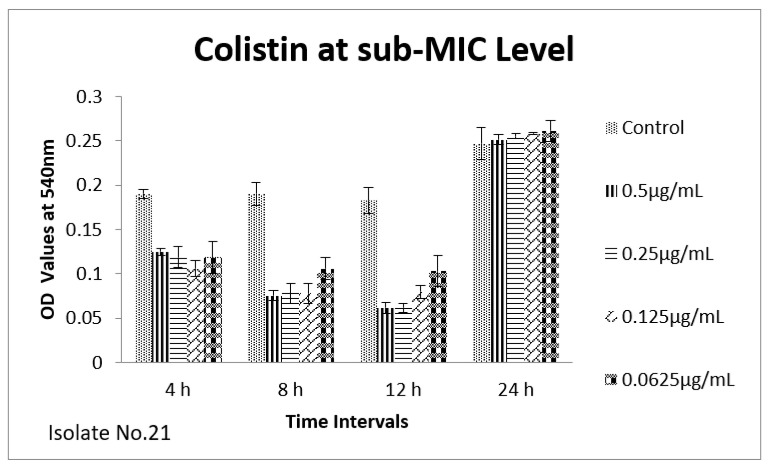
Shown are the effects of colistin on *E. coli* biofilm at different time intervals (OD 540 nm). Bar graph represents highest biofilm formation after 24 h of incubation.

**Table 1 antibiotics-11-01663-t001:** Comparison of antibiotic resistant traits of MDR and XDR phenotypes of *E. coli* isolates (*n* = 83). *p* values were calculated by chi-square test.

Resistance Traits	Total Isolates *n* (%)	MDR (*n* = 70)	XDR (*n* = 12)	*p* Value
Ampicillin	80 (96.38)	67	12	0.4650
Amoxicillin	75 (90.36)	62	12	0.1179
Levofloxacin	51 (61.44)	40	10	0.0857
Ciprofloxacin	51 (61.44)	42	8	0.6618
Tobramycin	5 (6.02)	2	3	0.0031
Gentamycin	7 (8.43)	3	4	0.0009
Neomycin	44 (53.01)	32	12	0.0005
Streptomycin	67 (80.72)	55	11	0.2902
Tigecycline	0	0	0	-
Tetracycline	75 (90.36)	65	9	0.0541
Oxytetracycline	80 (90.38)	68	11	0.3505
Doxycycline	50 (60.24)	40	10	0.0857
Nitrofurantoin	7 (8.43)	4	3	0.0272
Chloramphenicol	53 (63.85)	42	11	0.0340
Cefotaxime	1 (1.20)	1	0	0.6770
Cefixime	6 (7.22)	2	4	0.0002
Cephalothin	36 (43.37)	26	10	0.0029
Ceftazidime	6 (7.22)	4	2	0.1783
Cefepime	4 (4.81)	1	3	0.0005
Imipenem	4 (4.81)	2	2	0.0402
Meropenem	8 (9.63)	5	3	0.0541
TMP-SMX	64 (77.10)	52	11	0.1873
Lincomycin	83 (100)	70	12	0.1945
Augmentin	4 (4.81)	2	10	0.0001
Polymyxin B	1 (1.20)	1	0	0.6770
Colistin (Polymyxin E)	22 (26.50)	16	6	0.0499
**Virulence Genes**				
*fimH*	32 (38.55)	25	6	0.3457
*papC*	21 (25.30)	19	2	0.4424
*iutA*	34 (40.96)	26	7	0.1667
*kpsMT-II*	23 (27.71	18	4	0.5821
*papEF*	9 (10.84)	6	3	0.0925
*papGII*	22 (26.50)	17	5	0.2093
*fyuA*	13 (15.66)	10	3	0.3478
*papGIII*	0	0	0	-
**Resistance Genes**				
*bla* _TEM-1_	22 (26.50)	17	5	0.2093
*bla_OXA_*	8 (9.63)	4	4	0.0029
*bla_SHV_*	4 (4.81)	1	3	0.0005
*bla_PSE_*	0	0	0	-

MDR, Multi-drug Resistant; XDR, Extensively Drug Resistant.

**Table 2 antibiotics-11-01663-t002:** Comparison of antibiotic resistant traits of ESBL producing *E. coli* isolates (*n* = 83). *p* values were calculated by chi-square test.

Resistance Traits	Total Isolates *n* = 83 (%)	ESBL Producers *n* (%)	Non-ESBL *n* (%)	*p* Value
Ampicillin	80 (96.38)	38 (47.50)	42 (52.5)	0.5142
Amoxicillin	75 (90.36)	35 (46.66)	40 (53.33)	0.3942
Levofloxacin	51 (61.44)	26 (50.98)	25 (49.01)	0.5211
Ciprofloxacin	51 (61.44)	31 (60.78)	20 (39.2)	0.0038
Tobramycin	5 (6.02)	3 (60)	2 (40)	0.5857
Gentamycin	7 (8.43)	4 (57.14)	3 (42.85)	0.6204
Neomycin	44 (53.01)	27 (61.36)	17 (38.63)	0.0108
Streptomycin	67 (80.72)	32 (47.76)	35 (52.23)	0.8721
Tigecycline	0	0	0	-
Tetracycline	75 (90.36)	38 (50.66)	37 (49.33)	0.1673
Oxytetracycline	80 (90.38)	38 (47.5)	42 (52.5)	0.5142
Doxycycline	50 (60.24)	25 (50)	25 (50)	0.6850
Nitrofurantoin	7 (8.43)	4 (57.1)	3 (42.85)	0.6204
Chloramphenicol	53 (63.85)	28 (52.83)	25 (47.16)	0.2611
Cefotaxime	1 (1.20)	1 (100)	0	0.2969
Cefixime	6 (7.22)	6 (100)	0	0.0084
Cephalothin	36 (43.37)	22 (61.11)	14 (38.88)	0.0393
Ceftazidime	6 (7.22)	4 (60)	2 (40)	0.5804
Cefepime	4 (4.81)	4 (100)	0	0.0335
Imipenem	4 (4.81)	3 (75)	1 (25)	0.2714
Meropenem	8 (9.63)	6 (75)	2 (25)	0.1104
TMP-SMX	64 (77.10)	32 (50)	32 (50)	0.5453
Lincomycin	83 (100)	40 (48.19)	43 (51.80)	>0.9999
Augmentin	4 (4.81)	3 (75)	1 (25)	0.2714
Polymyxin B	1 (1)	0	1 (100)	0.2969
Colistin (Polymyxin E)	22 (26.50)	13 (59.09)	9 (40.90)	0.2327
*fimH*	32 (38.55)	17 (53.12)	15 (46.87)	0.4763
*papC*	21 (25.30)	8 (38.09)	13 (61.90)	0.2840
*iutA*	34 (40.96)	16 (47.05)	18 (52.94)	0.0039
*kpsMT-II*	23 (27.71	7 (30.43)	16 (69.56)	0.0450
*papEF*	9 (10.84)	5 (55.55)	4 (44.44)	0.6397
*papGII*	22 (26.50)	11(50)	11(50)	0.8431
*fyuA*	13 (15.66)	10 (76.92)	3 (23.07)	0.0240
*papGIII*	0	0	0	-
*bla* _TEM-1_	22 (26.50)	22 (100)	0	0.0001
*bla_OXA_*	8 (9.63)	8 (100)	0	0.0020
*bla_SHV_*	4 (4.81)	4 (100)	0	0.0335
*bla_PSE_*	0	0	0	-

ESBLs, Extended Spectrum β-lactamases.

**Table 3 antibiotics-11-01663-t003:** MIC-p and MIC-b analysis of selected antibiotics.

Isolate No.	MIC-p µg/mL				MIC-b µg/mL			
	GEN	COL	CEF	ENR	GEN	COL	CEF	ENR
7	4	2	0.5	0.06	1	1	16	4
21	1	2	0.5	0.125	1	1	32	4

MIC-p, Minimum Inhibitory Concentration for planktonic bacteria; MIC-b, Minimum Inhibitory concentration for biofilm; GEN, Gentamicin; COL, Colistin; CEF, Ceftriaxone; ENR, Enrofloxacin.

**Table 4 antibiotics-11-01663-t004:** Analysis of * MBIC/MRC and MBEC of selected antibiotics on bacterial biofilms.

Isolate No.	GEN		COL		CEF		ENR	
	MBIC µg/mL	MBEC µg/mL	MBIC µg/mL	MBEC µg/mL	MBIC µg/mL	MBEC µg/mL	MBIC µg/mL	MBEC µg/mL
7	8	32	8	128	128	>2048	256	>2048
21	32	128	16	64	256	>2048	64	>2048

MBIC/MRC, Minimum Biofilm Inhibitory Concentration; MBEC, Minimum Biofilm Eradication Concentration; GEN, Gentamicin; COL, Colistin; CEF, Ceftriaxone; ENR, Enrofloxacin. * Term MBIC/MRC is used synonymously. * MBIC/MRC is used synonymously.

## Data Availability

Data and strains used in this study can be made available upon request.

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
