# Peer review of "Investigating the Role of Antibiotics on Induction, Inhibition and Eradication of Biofilms of Poultry Associated Escherichia coli Isolated from Retail Chicken Meat"

_antibiotics, 2022, doi:10.3390/antibiotics11111663_

Round 1

Reviewer 1 Report

This manuscript “Investigating the role of antibiotics on induction, inhibition and eradication of biofilms of poultry associated Escherichia coli isolated from retail chicken meat” by Noreen et al. studied the antibiotic resistance profile of E. coli isolated from poultry sources. The study of the relationship between biofilm formation and colistin is appreciated, and revisions clarifying the importance of the findings will be needed.

Major:

1.     Please clarify all statistical analyses used in this study.

2.     Please check the graphs aligned in this study.

3.     Method 4.2 any replication for the tests?

4.     Line 293: what is the percentage of crystal violet used?

Minor

1.     Please double-check the language throughout the manuscript.

2.     Please be sure all genes are italicized throughout the manuscript.

3.     Line 13: “bacterial” antibiotic resistance

4.     Please check the language and format in method 4.3

Author Response

Introduction:  Improved as suggested

Redundancy in references:  addressed and numbers of references were reduced.

Method:  improved for language, format and clarity.

Results: improved for more clarity.

Conclusion: adjusted according to the findings.

 Major

  1. Comment: Please clarify all statistical analyses used in this study.

Correction: Statistical analyses along with names of applied test and p values were incorporated. Figures were double checked. 

  1. Comment: Please check the graphs aligned in this study.

Correction: Graphs aligned and modified for clarity, incorporated as tiff tag image files.

  1. Comment: Method 4.2 any replication for the test.

Correction: Phenotypic detection of ESBL was aligned with genotypic detection, since all the phenotypic ESBL producers were verified for the presence of ESBL genes listed in the study. Antibiotic susceptibility testing was repeated twice. In addition, MIC-p for the colistin, ceftriaxone, enrofloxacin and gentamicin were measured that confirmed the sensitivity data of disk diffusion testing for all the isolates. Since it was not convenient to identify many isolates sensitive to all four listed antibiotics simultaneously, to be sure, we tested our entire collection for MICp determination for all four antibiotics (colistin, ceftriaxone, enrofloxacin and gentamicin) and selected sensitive strains for biofilm formation assays. Our MIC data aligned perfectly with our disc diffusion data.

  1. Comment: Percentage of crystal violet used.

Correction: was 2%, now it’s mentioned in the text.     

Minor

  1. Language was double checked, improvements were made.
  2. All genes are italicized and in proper format throughout the manuscript.
  3. Line 13. Corrected
  4. Language and format errors removed in method 4.3

Reviewer 2 Report

General comment:

The paper is presenting the role of antibiotics on the inhibition and eradication of biofilms associated with E. coli isolated from retail chicken market. The design of the study is attractive, the results could have been interpreted in a way that would of increased the value of the paper.

Specific comments:

1. Introduction is presenting a lot of futile information and data that is known for decades.

2. Materials and methods are presenting methods that have been performed before and do not require detailed description.

More than two strains should be selected for reliable and valuable results.

MDR and XDR were not defined. For MDR is it 3, 5 or 7 antibiotics?

3. In the results section, for tobramycin and nitrofurantoin  I do not understand why only 5 and 7 strains respectively were tested to these antimicrobials.

4. The discussion section is well documented and represents the best part of the study.

Author Response

Introduction:  Improved as suggested, gives a clearer background of the study.

Redundancy in references:  addressed and numbers of references were reduced.

Method:  improved for language, format and clarity.

Results: improved for more clarity.

Conclusion: adjusted according to the findings.

Specific comments

Comment: Introduction is presenting a lot of futile information and data that is known for decades.

Correction: Introduction was made concise and unnecessary information was removed.

Comment: Materials and methods are presenting methods that have been performed before and do not require detailed description.

Correction:  Method section was amended and made clearer unnecessary details were removed.

Comment: more than two strains should be selected for reliable and valuable results.

Correction: Since it is difficult to identify many strong biofilm former isolates, sensitive to all four listed antibiotics simultaneously; to be sure, we tested our entire collection for MICp determination for all four antibiotics (colistin, ceftriaxone, enrofloxacin and gentamicin). We selected only sensitive and strong biofilm former strains for studying the effects of antibiotics. This is the strongest point of this study in comparison to other studies.

Comment: MDR and XDR were not defined.

Correction: Reference is now included please see reference number 37.

Comment: In the result section for tobramycin and nitrofurantoin, I do not understand why only 5 and 7 strains respectively were tested to these antimicrobials.

Correction: In fact all isolates were tested for both antibiotics only n=5 and n=7 isolates were found resistant. Described are actually the numbers not the names of the strains. 

Round 2

Reviewer 1 Report

Thank you for the changes.